# Explicit and Implicit Measures of Black Cat Bias in Cat and Dog People

**DOI:** 10.3390/ani14233372

**Published:** 2024-11-22

**Authors:** Greg C Elvers, Brianna Gavin, Robert J. Crutcher

**Affiliations:** Department of Psychology, University of Dayton, Dayton, OH 45469-1430, USA; gavinb1@udayton.edu (B.G.); rcrutcher1@udayton.edu (R.J.C.)

**Keywords:** black cat bias, implicit vs. explicit measures, cat vs. dog people

## Abstract

In some cultures, black cats are viewed as causing bad luck. This may partially explain why it takes longer for black cats to be adopted from animal shelters. Three possible causes of this bias against black cats—superstitious beliefs, belief in witchcraft, and religiosity—were investigated in people who self-identified as liking dogs more than cats, cats more than dogs, and liking dogs and cats equally. Superstitious beliefs did not predict black cat bias. Religiosity and belief in witchcraft predicted black cat bias but only for people who reported liking dogs more than cats. In a second study, black cat bias was measured in the same people twice—at least six weeks before Halloween, a holiday associated with superstition and witchcraft, and no more than three weeks before Halloween. Black cat bias was stronger when measured right before Halloween. This suggests that black cat bias might be modified by environmental factors.

## 1. Introduction

The history of human interactions with cats, especially black cats, is filled with folklore and superstitions. While a common superstition in the United States is that black cats bring bad luck, that is not universally true across cultures and times. Across cultures, there are many superstitions, both good and bad, about black cats [1]. For example, until 1975, black cats were required to be onboard British ships as a good luck omen [1]. In southern England, it was once believed that owning a black cat would lead to the marriage of the person’s daughters and that a black cat was considered a lucky wedding gift [2]. While unfortunate for the cat, some folklore suggests that supernatural powers can be derived from black cats. In parts of the Middle East, it was once believed that eating black cat’s meat protected a person from magic [1]. In Ireland, it was believed that killing a black cat, placing a bean in its heart, and burying it would result in magical powers, such as invisibility, to anyone who ate a bean that grew from such a plant [1].

Some of the negative superstitions associated with black cats may arise from the belief that black cats are associated with witchcraft and heresy [1]. Pope Gregory IX issued the papal bull (an edict from the Pope) *Vox in Rama* in 1233, and some believed that it proclaimed that cats were the vessels of the devil [3]. According to some authors, this led to the torture and death of many cats, especially black ones [3]. Negative attitudes toward black cats in the United States are evident from the quotations collected by Wayland Hand of the Center for the Study of Comparative Folklore and Mythology at the University of California, Los Angeles. Examples include the following quotations: “It is bad luck to see a black cat”; “If you see a black cat at night, walk away, or you’ll have bad luck”; “If you see a black cat, spit three times at it and you won’t have bad luck”; “If a black cat comes to your door, bad luck. The blacker the cat, the blacker the luck” [4].

While some people might still believe in such superstitions, sometimes having a black or dark coat can have its own consequences for a cat. In the United States, cats with black or brown coats are less likely to be adopted than cats with other coat colors [5]. After looking at adoption records of over 29,000 cats from two shelters to see if coat color (black only, black with other colors, and other) influenced adoption, black cats took about 2.25 days longer to be adopted, on average, than cats who were not primarily black, a statistically reliable difference [6]. At another shelter, black kittens took about 4 days longer to be adopted than kittens who were not primarily black, while black adult cats took almost six days longer, on average, to be adopted [6]. Brown and Morgan found that lighter-colored, younger cats had a shorter length of stay in a shelter than older, more darkly colored cats, but the greatest length of stay was for cats with yellow coats [7]. Similar results were found in the Czech Republic, where a study that looked at adoption records of 2170 cats from three cat shelters found that cats with darker, not necessarily black, coats stayed in the shelter approximately one month longer than cats with lighter coats [8]. Not all research finds that black cats stay longer in the shelter: one study found that white cats took longer to be adopted than black cats [9].

The observation that black cats take longer to be adopted [5,6,7,8] in some shelters could be partially due to black cat bias, or it could be that there are more cats in shelters with black coats than with other coat colors. There are multiple reasons why there might be more cats with black coats in a shelter. First, it could be that there are more black cats in the population, and, thus, more black cats find their way to shelters, and the length of stay has nothing to do with black cat bias. Second, if coat colors are approximately equally represented in the population, more black cats could find their way into shelters because they are viewed as less desirable and are, therefore, surrendered to shelters more often than cats with other coat colors. Thus, black cat bias could influence the number of black cats in shelters, which, in turn, might influence the length of stay. Third, even if initially there are fewer black cats in a shelter than cats with other coat colors, if people are biased against black cats and they are less likely to be adopted, eventually, the number of black cats relative to other coat colors in the shelter will rise. Thus, the base rate of black cats in a shelter that has been open for a long time cannot by itself be used as an argument against black cat bias unless the base rate of coat colors is controlled.

What is needed is a metric that considers the base rate of cats with various coat colors when looking at the length of stay in a shelter. The odds ratio (OR) is such a metric. In this context, an odds ratio would calculate the ratio of the number of black cats with a longer length of stay in a shelter to the number of black cats with a shorter stay. As long as the ratio stays the same, the absolute number of black cats in the shelter does not influence the calculation. The odds ratio also calculates the ratio of the number of non-black cats with a longer length of stay to the number of non-black cats with a shorter length of stay. Again, as long as the ratio stays the same, the absolute number of non-black cats does not influence the calculation. Finally, the odds ratio is defined as the first ratio (for black cats) divided by the second ratio (for non-black cats). Lepper et al. looked at the odds ratio for coat color and length of stay [5]. In their study, an OR above 1 indicates that cats of a given coat color are adopted more quickly than tabby cats, while an OR below 1 indicates that cats of a given coat color are adopted more slowly than tabby cats. They found that white (OR = 1.61), color point (OR = 1.26), and gray (OR = 1.25) cats were more likely to be adopted than cats with tabby color, and that brown (OR = 0.56) and black cats (OR = 0.59) were less likely to be adopted than tabby-colored cats [5].

While the greater length of stay for cats with darker coats in some shelters may not seem like much, even a few extra days at a shelter can have a serious, even fatal, impact on the cat—the risk of contracting an infection depends on the length of the stay at the shelter [10,11]. At the time of this research, the shelter in the United States studied in [5] euthanized animals that were not quickly adopted; the prevalence of “kill” shelters varies by country.

People think cats with various coat colors have different personalities [12]. One hundred eighty-nine participants rated cats with five different coat colors (bi-colored, black, orange, tri-colored, and white) on ten different terms (e.g., friendly, shy, and tolerant). People perceived black cats as more active and bolder than white cats, more aloof and shy than orange cats, and more friendly and tolerant than tri-colored cats [12]. Another study of 2822 cats found that black cats were perceived as more energetic and stubborn than cats with other coat colors and patterns [13]. While the perceptions may or may not reflect the actual behaviors of cats, they point to one possible explanation of why black cats take longer to be adopted: differences in perceived personalities. However, this list of perceived traits of black cats includes some traits that many people would likely find positive, and satisfaction with a cat is directly correlated with the perceived friendliness of the cat [14]. Some people might view a tolerant cat as more desirable. Perhaps traits that some people perceive as negative, such as aloofness and shyness, carry more importance in a person’s mind than more positive traits, and it is these negative traits that lead to negative feelings about black cats.

Similar to “big black dog syndrome”, which claims that large, black dogs take longer to be adopted and have higher rates of euthanasia [15,16,17,18], Jones and Hart [19] used the term “black cat bias” to describe the negative behaviors and attitudes toward black cats. They studied the prevalence and predictors of black cat bias by having participants answer questionnaires on superstitious behaviors, religiosity, and racial attitudes. Their participants also rated pictures of black and non-black cats on the perceived aggressiveness and friendliness of the cats, how well they thought they could read the cat’s emotions, and their willingness to adopt the cat [19]. Differences in ratings for non-black and black cats indicated the strength of black cat bias, if any. Jones and Hart found evidence of black cat bias. Superstitious beliefs and beliefs about how well the participants thought that they could read the emotions of the cats predicted black cat bias. Religiosity and racial attitudes did not predict black cat bias.

Explicit ratings, directly asking for people’s attitudes, such as Jones and Hart did [19], can provide useful information in many situations. However, when dealing with prejudice and bias, explicit measures can sometimes yield answers that are consistent with cultural expectations rather than accurately reflecting the participant’s true attitudes [20]. For example, when asking people whether they are prejudiced against a particular group, people may give the culturally expected answer of “no” even if they have strong attitudes against the group. Because half of the rated pictures of cats in Jones and Hart were of black cats, it is possible that some participants determined the purpose of their study and followed the stereotypical bias toward black cats when rating the cats. Thus, it is possible that Jones and Hart underestimated the extent of bias toward black cats.

An alternative to asking people explicitly about their attitudes is to measure their attitudes with an implicit associations test (IAT). The IAT assumes that if two concepts are strongly related to each other in a person’s thoughts, this association can influence a person’s behaviors. The article that introduced the IAT [20] has been cited nearly 7000 times according to the Social Sciences Citation Index, which makes it one of the most influential developments in psychology in the last few decades [21]. While the IAT is commonly used in psychological research, some researchers have questioned its reliability and validity [22]. For example, the IAT may be affected by factors other than the association between two concepts [23]. When a person is highly familiar with the stimuli used to test the association between concepts, familiarity influences the results of the IAT [23]. The authors also note that even when people have low familiarity with the stimuli used to test the concepts, the IAT retains some sensitivity for measuring the association between the concepts [23].

In the Brief IAT, people memorize two categories and then sort exemplars and non-exemplars of the categories while their reaction time is measured [24]. For example, people might memorize the two categories, black cats and bad words. The more strongly the two categories are related to each other in the person’s mind, the more quickly they should be able to sort exemplars of the categories from non-exemplars of the categories. They would then see a series of pictures of black and non-black cats and bad (e.g., hate) and good (e.g., love) words. If a picture of a black cat or a bad word (an exemplar of either of the two categories) appears, they would press one key; all other pictures (non-black cats) and words (good words) would require a different key press. The task would then be repeated with the first category changed. For example, the two categories might be non-black cats and bad words. If these two categories are less strongly related in the person’s mind, the sorting of exemplars should take longer than sorting the original categories. The difference in the reaction times of the two tasks is divided by the standard deviation of the reaction times, yielding a measure called *d*. If the association between one pair of categories is stronger than between the other pair of categories, *d* should differ from zero. For example, if black cat bias exists, black cats and bad words should be more strongly associated within the person than non-black cats and bad words. In that case, the reaction time to sort black cats and bad words should be lower (quicker responses) than to sort non-black cats and bad words. Responding in a culturally appropriate way, unless such a response matches the participant’s actual attitudes, may be much more difficult with an implicit measure than with most explicit measures [20].

It is possible that the extent, if any, of black cat bias is different for different groups. For example, people who self-identify as liking cats, “cat people”, may have more favorable opinions of cats in general and be less prone to black cat bias in particular than people who self-identify as liking dogs, “dog people”. Cat people who have owned cats may have had more positive experiences with cats, while dog people who have not owned cats may have had more experiences with neighborhood or friend’s cats that they perceive as negative or fewer experiences with cats in general. These experiences and preconceived notions about cats might lead to more or less black cat bias. People who have not been repeatedly exposed to cats as pets might lack an understanding of cat behavior and might default to the stereotypical idea, at least in some cultures, that black cats bring bad luck and, therefore, exhibit a greater bias against black cats.

Online profiles of pets available for adoption that were written in an analytic style led to faster adoptions than profiles written with peripheral cues (such as photographs of the animal) and social words (humanizing details about the animals) [25]. Photographs might influence how willing a person is to adopt a pet because they might lead the person to use a cognitive shortcut based on the color and visual appeal of the pet instead of more relevant features of the animal [25]. Thus, one quick and easy way of possibly reducing black cat bias might be to make people attend less to the coat color of the cat and more to the behavior of the cat. This might be accomplished by providing a description of the cat’s behavior and/or personality.

The following studies measured superstitious beliefs, beliefs in witchcraft, religiosity (primarily Christianity), and whether a person self-identified as a cat person, a dog person, a dog-and-cat person, or neither-a-dog-nor-a-cat person. The studies used both explicit and implicit measures of black cat bias. In the first study, it is predicted that participants who saw only a picture of the face of the cat they were rating would show more black cat bias than participants who saw both a picture of the face of the cat and a description of the cat. It is predicted that black cat bias will be found with both explicit and implicit measures and will be stronger for people who self-identify as dog people than cat people. It is predicted that black cat bias will be directly related to beliefs in witchcraft, superstitious beliefs, and religiosity and that these relations will be stronger in dog people than cat people.

## 2. Materials and Methods

### 2.1. Study One

The first study addressed possible correlates of black cat bias in self-identified cat people, dog people, dog-and-cat people, and people who liked neither cats nor dogs. Both explicit and implicit measures of black cat bias were used.

#### 2.1.1. Sample

Participants consisted of 71 females and 43 males who volunteered through a website designed for the recruitment of participants for online studies (https://www.prolific.com, accessed on 1 October 2023). Participants received USD 3.00 for their time. Table 1 shows the number of females and males who self-identified as a cat person, a dog person, a dog-and-cat person, and neither-a-dog-nor-a-cat person. We recruited only participants who reported that they were currently living in the United States of America and who were native English speakers. The participants had a mean age of 35.4 years (SD = 11.6 years). This study was approved by the Research Review and Ethics Committee at the University of Dayton.

#### 2.1.2. Materials

Participants completed a brief demographics questionnaire that asked for their age, sex, and the number of cats they currently lived with. The questionnaire also had a single question that asked if the participants viewed themselves as a cat person, a dog person, a dog-and-cat person, or neither-a-dog-nor-a-cat person.

The participants also completed the Belief in Paranormal Phenomena (BPP) scale, which is both reliable (test-retest *r* = 0.67) and valid (based on intercorrelations with seven other scales) [26]. The BPP consists of seven sub-scales: superstitious beliefs; witchcraft; religiosity (Christianity); Psi; spiritualism; extraordinary lifeforms; and precognition. The BPP consists of 25 questions (e.g., “The number 13 is unlucky” from the superstitious beliefs subscale) with 5-point Likert scale responses (1 = strongly disagree; 5 = strongly agree). The questions were presented one at a time, and the participant could not return to previous questions.

#### 2.1.3. Design

After giving consent, participants answered the demographic questionnaire and the BPP. Next, participants performed the explicit black cat bias task. This task consisted of looking at a picture of the face of a cat (see Figure 1 for sample pictures). The black cats had an entirely black coat with neither white nor orange, as might be found in tortoiseshell cats, calico cats, or cats with a spot of white. The non-black cats had no large black areas; they were grey, grey-striped, brown-striped, orange-and-white, or grey-and-white. Approximately half of the participants (*n* = 52) were also randomly assigned to read a description of the cat in addition to seeing the picture. A sample description is as follows:

Patches is an orange-and-white cat who is about two years old. Patches loves to play with toy mice and chase strings. Patches is very friendly and likes to be around people. When sleepy, Patches thinks that curling up on your lap is the best place to take a cat nap.

The participants then answered two questions with a 5-point Likert scale (1 = strongly disagree; 5 = strongly agree): (1) I would like to live with this cat and (2) This is a good cat. The participants completed this task for four black cats and for four non-black cats whose pictures were presented in a random order. For the participants who viewed descriptions in addition to pictures, a different random pairing of pictures and descriptions that were appropriately modified for the cat’s coat color and pattern were used for each participant.

Finally, participants performed the brief implicit association task (IAT). Following the general procedure for the brief IAT [24], participants viewed two categories (“black cats and bad words” or “non-black cats and bad words”). The good (happy, warm, love, and friend) and bad (angry, cold, hate, and enemy) words were taken from an example in [24]. The participants then saw 10 pictures of cats, both black and non-black, and 10 words, both bad and good, and sorted them into two categories. Participants pressed the “k” (for keep) key on a keyboard for pictures or words that were in either of the two categories. Participants pressed the “d” (for discard) key for pictures or words that were in neither category. For example, if the categories were black cats and bad words, the person should press the “k” key if they saw a black cat or one of the bad words (angry, cold, hate, enemy). If they saw a non-black cat or a good word (happy, warm, love, friend), they should press the “d” key. The reaction time from a picture or word onset to the correct key press was recorded. Participants were instructed to respond as quickly as they could, even if it meant making a few errors. If a participant made an error, a red X appeared, and the participant had to make the correct response. The pictures and words were presented in a block randomized order, with the exception that the first four stimuli were always from one of the two categories. These four trials were meant to establish the two categories in the participant’s mind and were not used in the data analysis.

Participants completed four blocks of the brief IAT. The first and last blocks were randomly selected to be either “black cats and bad words” or “non-black cats and bad words”. The middle two blocks used the other pair of categories. The blocks were counterbalanced with an ABBA design.

#### 2.1.4. Data Analysis

The BPP was scored following the method given in [26]. The implicit measure of black cat bias was created following the method given in [24]. The explicit measure of black cat bias was created by subtracting the mean responses to the questions, “I would like to live with this cat” and “This is a good cat” for black cats, from the corresponding mean responses for non-black cats.

For both explicit and implicit measures of black cat bias, one-tailed, one-sample *t*-tests were used to determine if black cat bias likely existed. To protect the family-wise α level, the Bonferroni correction was applied. To determine if superstitious behaviors, beliefs in witchcraft, and religiosity were the predictors of implicit and explicit measures of black cat bias, Pearson’s *r*’s were calculated. To determine if descriptions of cats reduced black cat bias, one-tailed, independent samples *t*-tests were used. A one-way, independent samples ANOVA with Tukey multiple comparisons was used to determine if the number of cats owned by cat people, dog people, and dog-and-cat people differed.

### 2.2. Study Two

Study two utilized a sample of university students with the expectation that they would consist largely of people who would self-identify as dog people—the people who, based on the results of the first study, are more likely to exhibit black cat bias. This expectation is based on the results of asking students in another class at the same university whether they were cat people, dog people, both dog-and-cat people, or neither-dog-nor-cat people. The primary purpose of study two was to see if black cat bias was stronger when measured near Halloween, which, in the United States, is a holiday associated with superstition and witchcraft, than when measured earlier in the year. Black cats are often portrayed in Halloween decorations as witches’ familiars. If belief in superstitions or witchcraft is a predictor of black cat bias, then black cat bias may be more extreme closer to Halloween because being exposed to such decorations around Halloween may reinforce people’s beliefs that black cats are associated with witches. If black cat bias changes around Halloween, this would indicate that it might be malleable and might be capable of being changed by external events.

#### 2.2.1. Sample

At time 1 (48 to 65 days before Halloween, which is observed on 31 October), 46 females and 8 males with a mean age of 18.9 years (SD = 0.9 years) participated in partial fulfillment of their research requirement in an introductory psychology class. At time 2 (0 to 17 days before Halloween), 33 of the females and 4 of the males returned to complete this study a second time. Only data from the 37 students who completed both sessions were used in the analysis. Twenty students self-identified as dog people, 13 identified as dog-and-cat people, 3 identified as cat people, and 1 identified as neither. This study was approved by the Research Review and Ethics Committee at the University of Dayton.

#### 2.2.2. Materials

The demographic questionnaire, BPP, and pictures from study one were also used in study two.

#### 2.2.3. Design

The design was identical to that in study one with the following exceptions: (1) descriptions were always present during the explicit cat rating task; (2) each participant participated twice (once at least 48 days before Halloween and once no more than 17 days before Halloween); and (3) a researcher was present while the students completed this study.

#### 2.2.4. Data Analysis

Implicit and explicit measures of black cat bias were scored as in study one. To determine if black cat bias existed far from and close to Halloween, one-tailed *t*-tests were used for both implicit and explicit measures of black cat bias.

## 3. Results

### 3.1. Study One

#### 3.1.1. Prevalence of Black Cat Bias

Black cat bias was expected with both the explicit and implicit measures. Self-identified cat people were expected to show less black cat bias than self-identified dog people. The mean responses to the questions, “I would like to live with this cat” and “This is a good cat” for black cats were subtracted from the corresponding mean responses for non-black cats. Positive values indicate that non-black cats were rated more highly than black cats and are an explicit measure of black cat bias. The implicit measure of black cat bias, *d*, is defined as the mean reaction time for responding to the “non-black cats and bad words” categories minus the mean reaction time for responding to the “black cats and bad words” categories. This difference is divided by the standard deviation of all reaction times. Positive values of *d* indicate the presence of black cat bias. That is, *d* will be positive if black cats and bad words are more strongly associated (therefore, having a faster reaction time while sorting) than non-black cats and bad words.

Table 2 summarizes the descriptive statistics and the results of one-tailed, one-sample *t*-tests comparing the values of the explicit measure of black cat bias to 0 for those who self-identified as cat people, as dog people, and as dog-and-cat people. Because only one person self-identified as neither-a-dog-nor-a-cat person, this category was not analyzed. To control for Type I errors across the family of tests, a Bonferroni adjusted α level of 0.05/3 = 0.017 was used. There was insufficient evidence of black cat bias when measured explicitly.

Table 2 also summarizes the descriptive statistics and the results of one-tailed, one-sample *t*-tests comparing the values of the implicit measure of black cat bias to 0. The implicit measure of black cat bias is statistically significant with a medium effect size (*r*^2^) [27] for people who self-identify as dog people and with a large effect size for people who self-identify as dog-and-cat people. The implicit measure of black cat bias is not statistically significant for the self-identified cat people and explains less than 4% of the variability in the data (a small effect size).

#### 3.1.2. Predictors of Black Cat Bias

Superstitious behaviors, belief in witchcraft, and religiosity were expected to be directly correlated with black cat bias measured both explicitly and implicitly. These relations are predicted to be stronger in dog people than in cat people.

At the Bonferroni corrected α level, superstitious behaviors were not a statistically reliable predictor of black cat bias with either the explicit or implicit measures for cat people (explicit: *r*(25) = −0.181, *p* = 0.188; implicit: *r*(25) = 0.241, *p* = 0.118), dog people (explicit: *r*(30) = −0.113, *p* = 0.273; implicit: *r*(30) = 0.174, *p* = 0.175), and dog-and-cat people (explicit: *r*(55) = −0.015, *p* = 0.457; implicit: *r*(55) = −0.231, *p* = 0.043).

As shown in Table 3, Pearson’s *r* revealed that belief in witchcraft was a predictor of explicitly measured black cat bias for dog people with a medium effect size but not for either cat people (at the Bonferroni corrected α level, but with a medium effect size) or dog-and-cat people with a small effect size. Fisher’s r to z transformation revealed that Pearson’s correlations between belief in witchcraft and black cat bias for cat people and dog people were reliably different; one-tailed *z* = 1.904; *p* = 0.029. As belief in witchcraft increases in dog people, black cat bias tends to increase. Belief in witchcraft was not a predictor of the implicit measure of black cat bias for dog people, cat people, or dog-and-cat people.

As shown in Table 4, Pearson’s *r* revealed that religiosity is a predictor of black cat bias for dog people when measured explicitly but not implicitly at the Bonferroni α level. As religiosity increases, black cat bias tends to increase for dog people when measured explicitly. Religiosity is not a predictor of either explicitly or implicitly measured black cat bias for cat people and dog-and-cat people.

Religiosity and belief in witchcraft are correlated with large effect sizes for cat people (*r*(25) = 0.761, *p* < 0.001), dog people (*r*(30) = 0.713, *p* < 0.001), and dog-and-cat people (*r*(55) = 0.580, *p* < 0.001).

#### 3.1.3. Reduction in Black Cat Bias

It was predicted that black cat bias would be reduced when a description was added to a picture of the cat. Because there was insufficient evidence of black cat bias when measured explicitly and only for dog people and dog-and-cat people when measured implicitly, only the implicit measure for dog people and dog-and-cat people was used with one-tailed independent-samples *t*-tests with a Bonferroni corrected α level of 0.05/2 = 0.025. The *t*-test failed to reveal a difference for the dog people (*M*_Descriptions_ = 0.228, *M*_No Descriptions_ = 0.053), *t*(29) = 1.924, *p* = 0.032, *r*^2^ = 0.113. While not statistically significant, this is a medium effect size in the direction *opposite* of the prediction. For dog-and-cat people, the *t*-test failed to reveal a difference (*M*_Descriptions_ = 0.173, *M*_No Descriptions_ = 0.109); *t*(29) = 1.012; *p* = 0.158; *r*^2^ = 0.034.

#### 3.1.4. Number of Cats Owned

A one-way, independent samples ANOVA revealed that the number of cats owned by self-identified cat people (*M* = 1.96), dog people (*M* = 0.29), and dog-and-cat people (*M* = 1.09) was different, *F*(2, 110) = 12.797, *MS*_error_ = 1.545, *p* < 0.001, $\eta_{p}^{2}$ = 0.189. Tukey multiple comparisons revealed a statistically reliable difference between each group, all *p* values < 0.013.

#### 3.1.5. BPP

Descriptive statistics for the Belief in Paranormal Phenomena scale are included in Appendix A for the first study and Appendix A for the second study.

### 3.2. Study Two

Similar to study one, one-tailed *t*-tests comparing the explicit measure of black cat bias to 0 failed to reveal black cat bias at time one (*M* = −0.030, *t*(36) = −0.399, *p* = 0.346, *r*^2^ = 0.004) and at time two (*M* = −0.068, *t*(36) = −1.172, *p* = 0.125, *r*^2^ = 0.037). The implicit measure of black cat bias was statistically reliable at both time one (*M* = 0.169, *t*(36) = 4.719, *p* < 0.001, *r*^2^ = 0.382) and time two (*M* = 0.238, *t*(36) = 7.839, *p* < 0.001, *r*^2^ = 0.630). The implicit measure of black cat bias was about 40% larger when measured close to Halloween. This is a statistically reliable difference with a small effect size, *t*(36) = 1.826, *p* = 0.038, *r*^2^ = 0.085. Being exposed to a holiday associated with witchcraft and superstition may increase black cat bias.

## 4. Discussion

In the first study, black cat bias was found for people who reported they were living in the United States, but only with the implicit measure and only for dog people and dog-and-cat people. In the second study, the implicit measure of black cat bias was found to be approximately 40% larger when measured closer to Halloween than farther from Halloween. Jones and Hart [19] found black cat bias with their explicit measures. The difference between the results from the two studies may be related to the questions asked in the explicit task. We intentionally avoided most of the explicit questions used by Jones and Hart. Jones and Hart had participants rate the perceived friendliness and aggressiveness of the cat based on a picture of the cat, which was taken from the websites of adoption agencies. We thought it unlikely that adoption agencies would post pictures of cats that were openly showing signs of fight vs. flight (e.g., piloerection, pinnae pulled back, crouched posture), and some of the signs of friendliness (e.g., rubbing/bunting, tail up, purring) are not easily captured in a still photograph of only the cat’s face. We also intentionally avoided a question on the perceived ability to read the emotions of the cat as cats tend to have a restricted set of emotions relative to humans—cats do not have emotions related to jealousy, pride, empathy, or guilt [28]—and emotional expression may not be well captured in a picture—blinking, hissing, nose-licking, and head turn bias are some correlates of the emotional state of cats [29], which may not be easily represented in a picture. Perhaps, it is exactly these attributes that are not well-captured in a picture that allow a person to express a bias toward black cats. That is, if an attribute of the cat is objectively identifiable in a picture, then cats of all coat colors should be treated the same, and the subjective bias against black cats might not be expressed.

There are several possible reasons why cat people and dog people might have differences in their bias toward black cats. Because self-identified cat people have, on average, more cats than dog people, cat people may have more experience with cats than dog people, and these experiences may offer protection from, or at least a reduction in, black cat bias. Alternatively, self-identified dog people may favor certain attributes of dogs over cats (otherwise, they might self-identify as a dog-and-cat person), and this preference for these attributes in dogs may lead to bias against black cats. If the first explanation (protection from black cat bias based on experience with cats) is true, one might expect that a self-identified dog-and-cat person should receive at least some protection from black cat bias from their liking of cats. If the second explanation (liking certain attributes of dogs leads to black cat bias) is true, self-identified dog-and-cat people might be subject to black cat bias. The results are consistent with the second explanation—cat people showed little black cat bias (small effect size), while both dog people (medium effect size) and people who identified as dog-and-cat people (large effect size) exhibited black cat bias when measured implicitly.

Explicitly measured black cat bias can be predicted from a belief in witchcraft in dog people—as belief in witchcraft increases, black cat bias also tends to increase. Black cat bias in cat people was not reliably related to their belief in witchcraft. Self-identified cat people own more cats and, thus, are likely to have more opportunities to see that beliefs about witches and cats are not true. This may make self-identified cat people less likely to be influenced by folklore about cats than dog people. This could be tested in future studies by asking dog-and-cat people how strongly they agree with folklore that involves black cats. Given the correlation between religiosity and belief in witchcraft, it is not surprising that black cat bias can also be predicted from dog people’s religious point of view. As religiosity increases for dog people, black cat bias also tends to increase. However, this effect was not reliable for cat people.

Partially consistent with the prediction, people who self-identify as a dog person or as a dog-and-cat person had negative attitudes toward black cats when measured implicitly. This bias is true in the sample drawn from the general population of people living in the United States and in students in an introductory psychology class at a university in the United States. The explicit measure of black cat bias failed to reveal a bias against black cats. Partially consistent with the predictions, the bias is more extreme around Halloween, which, in the United States, is a holiday associated with superstition and witchcraft, than a month and a half earlier when measured implicitly but not when measured explicitly. Contrary to the prediction, including a description of the cat with the picture did not reduce black cat bias. While not statistically significant, the trend was in the opposite direction.

If cat people exhibit little black cat bias, there must be some other explanation for findings that black cats take longer to be adopted from shelters [5,6,7]. One explanation is that black cat bias might arise because some people find black cats less attractive than cats with other coat colors. Another explanation is that perhaps black cats are physically harder to see than non-black cats when they are housed in small, possibly ill-lit cages in a shelter, and potential adopters simply do not see the cat in the cage. Extra lighting, especially in the back of a cage, or the addition of a brightly colored break-away collar might make black cats easier to see and reduce the time to adoption. Alternatively, one might look at shelter records to compare the length of stay in the shelter for black cats housed in cages versus black cats housed in an open room, which might be better lit. Another possible explanation is that the additional length of stay in the shelter might be due to people who self-identify as dog-and-cat people. These people exhibited black-cat bias similar to dog people, and if they were looking for a cat, black-cat bias might influence their decision about whether to adopt a black cat or a non-black cat.

Jones and Hart [19] failed to find that religiosity predicted explicitly measured black cat bias, while we found that it predicted explicitly measured black cat bias for dog people but not cat people. These differences could be due to differences in the questions used to explicitly measure black cat bias. One explanation is that Jones and Hart’s sample might have included a larger proportion of cat people than our sample, and the lack of bias in cat people washed out the effect in the larger group. Another possible explanation is that higher perceived aggressiveness and lower perceived friendliness—attributes measured by Jones and Hart but not us—are major causes of black cat bias.

The religiosity scale on the Belief in Paranormal Phenomena scale [26] is based on Christian beliefs. It would be interesting to see if other faith traditions are also associated with black cat bias. For example, Muslims are said to admire cats for their cleanliness and for being revered by the prophet Muhammad [30] (p. 131). Perhaps, this positive view of cats would lead to less black cat bias in Muslims than in Christians.

Bias against black cats also appears to be more extreme right before Halloween than one and a half to two months earlier. This finding is important because it suggests that black cat bias might be malleable—it might be changed by external factors. Future research could look at whether an intervention designed to reduce belief in witchcraft could influence black cat bias in dog people and whether the results of such an intervention would be the same for explicit and implicit measures. Future research could consider whether celebratory days such as World Cat Day, celebrated on 8 August, and National Black Cat Day, celebrated on 27 October, influence attitudes toward black cats in people who are aware and unaware of these days. This could further test the malleability of black cat bias in people who are aware of such days. Anything that reduces black cat bias might lead to shorter lengths of stay in shelters for black cats.

The finding that black cat bias might be larger closer to Halloween needs to be interpreted with caution due to the high attrition rate of the participants from time one to time two and due to the small sample of college students in an introductory psychology course that was used. It could be that being exposed to issues of diversity and social justice in coursework makes these students more sensitive to issues associated with bias, and they might not reflect the general population’s sensitivity to such issues. This finding must also be interpreted with caution because it was not feasible to counterbalance the time of this test—the close-to-Halloween test was always the second administration of this procedure.

While this study did not address issues about the length of stay in shelters for black and non-black cats, it suggests that black cat bias might be malleable. If black cat bias plays some role in their length of stay, shelters might be able to affect a change in attitudes toward black cats by not using any decorations, in particular, black cat decorations near Halloween.

No matter why black cats may or may not stay in shelters longer than cats with other coat colors, organizations such as The American Society for the Prevention of Cruelty to Animals (ASPCA) actively work with shelters to increase the adoption of black cats [31,32]. Their efforts and future research on ways of alleviating black cat bias may help ensure that black cats find loving homes in a timely manner.

## 5. Conclusions

In summary, the results indicate that black cat bias exists when measured implicitly for dog people and dog-and-cat people but not for cat people. When measured explicitly, belief in witchcraft is positively correlated with black cat bias in dog people. Black cat bias might be more extreme near Halloween than six to eight weeks earlier in the year. This result implies that black cat bias may be malleable and subject to external events.

## Figures and Tables

**Figure 1 animals-14-03372-f001:**
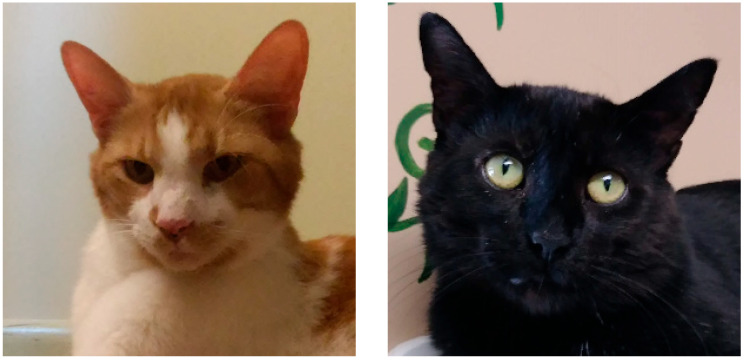
Examples of Pictures of Cats Used in Studies 1 and 2.

**Table 1 animals-14-03372-t001:** Number of Participants in Study 1 Who Self-Identified as a Dog-and-Cat Person, Cat Person, Dog Person, or Neither-a-Dog-nor-a-Cat Person.

	Dog-and-Cat Person	Cat Person	Dog Person	Neither	Total
Female	36 (31.6%)	14 (12.3%)	21 (18.4%)	0 (0.0%)	71 (62.3%)
Male	20 (17.5%)	12 (10.5%)	10 (8.8%)	1 (1.0%)	43 (37.8%)
Total	56 (49.1%)	26 (22.8%)	31 (27.2%)	1 (1.0%)	114

**Table 2 animals-14-03372-t002:** Prevalence of Black Cat Bias Measured Explicitly and Implicitly for People who Self-Identified as a Cat Person, a Dog Person, or a Dog-and-Cat Person.

	Explicit Measure	Implicit Measure
Cat People	*M* = 0.082	*M* = 0.056
	*SD* = 0.384	*SD* = 0.291
	*t*(25) = 1.085	*t*(25) = 0.982
	*p* = 0.144	*p* = 0.168
	*r*^2^ = 0.045	*r*^2^ = 0.037
Dog People	*M* = −0.024	*M* = 0.115
	*SD* = 0.317	*SD* = 0.253
	*t*(30) = −0.425	*t*(30) = 2.530
	*p* = 0.337	*p* = 0.008
	*r*^2^ = 0.006	*r*^2^ = 0.176
Dog-and-Cat People	*M* = −0.013	*M* = 0.145
	*SD* = 0.340	*SD* = 0.237
	*t*(55) = −0.295	*t*(55) = 4.575
	*p* = 0.385	*p* < 0.001
	*r*^2^ = 0.002	*r*^2^ = 0.276

Note. The explicit measure is the mean response to the ratings of non-black cats minus the mean response to the ratings of black cats. Positive values indicate black cat bias. The implicit measure is the difference in the reaction time for categorizing non-black cats and bad words vs. black cats and bad words. The difference is divided by the standard deviation of the reaction times. Positive values indicate black cat bias.

**Table 3 animals-14-03372-t003:** Belief in Witchcraft as a Predictor of Black Cat Bias Measured Explicitly and Implicitly for People who Self-Identified as a Cat Person, a Dog Person, or a Dog-and-Cat Person.

	Explicit Measure	Implicit Measure
Cat People	*r*(25) = −0.393	*r*(25) = 0082
	*p* = 0.023	*p* = 0.346
	*r*^2^ = 0.154	*r*^2^ = 0.007
Dog People	*r*(30) = 0.461	*r*(30) = 0.315
	*p* = 0.004	*p* = 0.042
	*r*^2^ = 0.213	*r*^2^ = 0.099
Dog-and-Cat People	*r*(55) = 0.207	*r*(55) = −0.097
	*p* = 0.063	*p* = 0.239
	*r*^2^ = 0.043	*r*^2^ < 0.001

**Table 4 animals-14-03372-t004:** Religiosity as a Predictor of Black Cat Bias Measured Explicitly and Implicitly for People who Self-Identified as a Cat Person, a Dog Person, or a Dog-and-Cat Person.

	Explicit Measure	Implicit Measure
Cat People	*r*(25) = −0.084	*r*(25) = 0.230
	*p* = 0.342	*p* = 0.129
	*r*^2^ = 0.007	*r*^2^ = 0.053
Dog People	*r*(30) = 0.479	*r*(30) = 0.347
	*p* = 0.003	*p* = 0.028
	*r*^2^ = 0.229	*r*^2^ = 0.120
Dog-and-Cat People	*r*(55) = 0.098	*r*(55) = 0.100
	*p* = 0.237	*p* = 0.231
	*r*^2^ = 0.010	*r*^2^ = 0.010

## Data Availability

Data are available from the first author upon request.

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
