# Peer review of "Explicit and Implicit Measures of Black Cat Bias in Cat and Dog People"

_animals, 2024, doi:10.3390/ani14233372_

Round 1
Reviewer 1 Report
Comments and Suggestions for Authors
see attached document

Author Response
- A central premise of this study is that black cats take longer to be adopted from animal shelters. The manuscript attributes this premise to one study (Leeper et al., 2002) more than 20 years old and a single study from the Czech Reuplic (Kubseova, et al. 2017) which is likely unrelated to the attitudes of American college students or the American public at large. Today, the notion that there are biases against black cats that can affect shelter adoptions is largely considered a “myth” within the sheltering community, as evidenced by reports on many shelter websites and veterinary and cat-related blogs (e.g. "Black Cats The Good, The Bad, and The Misunderstood". Las Vegas, Nevada: The Animal Foundation.; https://www.vetstreet.com/our-pet-experts/is-it-a-myth-that-black-shelter-pets-are-less- likely-to-be-adopted#). However, the myth is persistent and thus can explain potential impact on implicit biases, but is unlikely to be shown in actual behavior or measures of explicit bias- as is reported in this study.
Dr. Emily Weiss of the American Society for the Prevention of Cruelty to Animals looked at the data from the ASPCA’s Comprehensive Animal Risk Database, which pulled numbers from 14 communities and nearly 300,000 dogs and cats, to learn more (http://www.aspcapro.org/blog/2014/06/18/black-new-white). It all comes down to intake numbers, and there are more black dogs and cats in shelters than any other color. Thirty percent of the dogs taken in throughout 2013 were black, with brown coming in second at 23 percent. Black cats made up 33 percent of the feline intake, with gray cats coming in a distant second, at 22 percent. So, per the research, if three black dogs and one white dog enter a shelter, and one black dog and one white dog are adopted that day, that still leaves two black dogs waiting for homes. That means that the likelihood of your seeing more black dogs than any other color in that shelter is pretty high but it does not necessarily mean that black dogs are being overlooked because of their color or that they aren’t being adopted at all. The other reason Dr. Weiss suspects the belief persists is simply because we’re human, and as studies have shown, we’re sometimes more likely to cling to a belief when confronted with facts and evidence that prove it wrong.
Response: Thank you for this helpful comment. We have updated the paragraph in the manuscript that talks about adoptions of cats with various coat colors. The paragraph now includes a study with a different finding (white cats stay longer than black cats). We have also included a paragraph that states that the adoption rate of black cats might be influenced by the base rate of black cats in shelters. The paragraph also discusses how the base rate of black cats in shelters might be partially influenced by black cat bias. Finally, we have added an additional paragraph about a metric, the odds ratio, that takes into account the base rate of cats with various coat colors. Lepper et al. (2002) looked at the odds ratio for the adoption of cats of various coat colors and found that, relative to tabby color, black cats were less likely to be adopted (OR = 0.59). Cats with white, color point, and gray coats all were more likely to be adopted (ORs > 1) than tabby color. Because the odds ratio includes base rate information (in the table below, the OR is 6 whether there are fewer black cats than non-black cats or more black cats than non-black cats), Lepper’s results suggest that a black coat lowers the likelihood of being adopted independent of the number of black cats relative to non-black cats that are present, at least in their study.
|
|
Black |
Non-Black |
|
Black |
Non-Black |
|
Adopted soon |
5 |
30 |
|
500 |
30 |
|
Adopted late |
10 |
10 |
|
1000 |
10 |
While the anecdotal and more formal evidence provided on websites and blogs strongly suggest that part of black cat bias, as measured by length of stay in shelters, is based on the base rates of black and non-black cats, the current manuscript address other aspects of black cat bias that are independent of base rate information. That is, we did not look at length of stay as an independent or dependent variable in either study. Of course, additional scientific, peer-reviewed studies of black cat bias measured by length of stay that includes base rate information would be helpful.
- The discussion of people’s belief that coat color reflects personality (lines 80-93) are likely largely irrelevant since, as the authors note (lines 89-90) traits that people find positive are among the traits associated with black cats.
Response: Thank you for pointing this out. To clarify, some of the traits that people find positive (active, friendly) are associated with black cats, but some of the traits that people might find negative (aloof, shy, stubborn) are also associated with black cats. As the paragraph states, some of the negative traits might be more important in a person’s decision than the positive traits. If that is true, then we believe that the relations between coat color and personality are relevant to the discussion of why some people may have some negative opinions toward black cats.
- The methods sections does not clarify how people are asked to self-identify as cat-people, dog people or dog and cat people. The demographics questionnaire should be included as an appendix. Actual cat or dog ownership would understandably be a much better indicator of familiarity with actual cat behavior, which this study does show to be a factor in largely overcoming any biases.
Response: Thank you. We have modified this section of the materials section to clarify that the questionnaire included a single question that asked if participants viewed themselves as a cat-person, a dog-person, a dog-and-cat person, or neither a dog-nor-cat person. We used this subjective measure because more quantifiable measures, such as cat or dog ownership or the number of cats or dogs owned may include many factors other than the preference for cats or dogs. For example, a person might prefer cats but lives with a dog because of a spouse’s preference or, if they live in an area with high crime rate, because dogs offer more protection than cats. A person might continue to live with a cat that they strongly disklike but continue to care for it as a responsibility they took on when they adopted the cat. A person might really like cats but not be allowed to have one due to allergies, cost, or restrictions placed by a landlord.
- The bulk of the reportedly significant findings of bias (at least among dog people) is based on measures of implicit bias (e.g. Greenwald etal. 1998). This methodology has been subject to considerable criticism, which should at least be mentioned. For example, it has been interpreted as assessing familiarity,( e.g. Ottaway et al. ,2001)i or mere cultural knowledge irrespective of personal endorsement of that knowledge (Arkes and Tetlockm 2004)ii. This reliance upon awareness of cultural myths surrounding black cats would explain the findings that this study finds some evidence of implicit but not explicit bias.
Response: Thank you for catching this oversight on our part. We have added statements that there are controversies about the IAT’s reliability and validity and have cited Ottaway et al. which you kindly provided.
- Finally, the study suggests that black cat bias might be malleable by external events. It would be helpful to comment on the extensive measures being taken by animal shelters to overcome what is widely recognized as a myth of black cat issues, particularly near Halloween. For example, many shelters feature special black cat adoption days with low cost adoptions or have other events specifically celebrating black cats ( see https://www.aspca.org/news/black-cat-appreciation-day-friday and https://www.aspcapro.org/resource/5-ways-promote-black-cats.
Response: Thank for this helpful suggestion which we have incorporated into the manuscript.
i Ottaway, S. A.; Hayden, D. C.; Oakes, M. A. (2001). "Implicit attitudes and racism: Effects of word familiarity and frequency on the implicit association test". Social Cognition. 19 (2): 97 144.
ii Arkes, H. R.; Tetlock, P. E. (2004). ""Attributions of Implicit Prejudice, or "Would Jesse Jackson 'Fail' the Implicit Association Test?". Psychological Inquiry. 15 (4): 257 278.

Reviewer 2 Report
Comments and Suggestions for Authors
The subject of the article is interesting and fits in with current trends in the discussion on the welfare of companion animals.
I have some concerns about combining two species of companion animals, a dog and a cat, in the research group. After all, there can also be prejudices against a black dog, e.g. the Hound of the Baskervilles. There were also studies and scientific articles on this topic: "Black dog syndrome in animal shelters", 2014, Goleman et al.
Why was a control group (0) not created in the survey for the first task, e.g. people who do not have/do not like either a dog or a cat?
Why was the opinion of respondents not presented in the survey for the second task 48 and 17 days before, e.g. World Cat Day (August 8) in my opinion.
In my opinion, this would be a good, contrasting reference to Halloween.
I have no reservations about the statistical methods.
I miss the reference in the conclusions to the application of the obtained results in everyday practice.
Author Response
The subject of the article is interesting and fits in with current trends in the discussion on the welfare of companion animals.
- I have some concerns about combining two species of companion animals, a dog and a cat, in the research group. After all, there can also be prejudices against a black dog, e.g. the Hound of the Baskervilles. There were also studies and scientific articles on this topic: "Black dog syndrome in animal shelters", 2014, Goleman et al.
Response: We apologize for the misunderstanding. We did not combine two species – the study only addresses biases against black cats. We did look at people who responded that they prefer cats vs people who prefer dogs vs people who prefer both cats and dogs. We have clarified this in the Materials section for study one – section 2.1.2.
- Why was a control group (0) not created in the survey for the first task, e.g. people who do not have/do not like either a dog or a cat?
Response: Thank you for your comment. We apologize if we are not correctly understanding your concern. The participants were asked to categorize themselves into one of four categories in both studies: a dog-and-cat person, a cat-person, a dog-person, and neither dog-nor-cat-person. Because only one person in the sample for the first study categorized themselves as neither-a-dog-nor-cat-person, we could not include them in the statistical analysis. To help clarify this, we have changed “neither” to “neither a dog-nor-cat-person” in the description of the sample and in the title of Table 1. Because of space constraints, we left “Neither” by itself within the table. Hopefully this will not cause confusion given the table’s title. - Why was the opinion of respondents not presented in the survey for the second task 48 and 17 days before, e.g. World Cat Day (August 8) in my opinion.
In my opinion, this would be a good, contrasting reference to Halloween.
Response: This is a good suggestion for future research. Unfortunately, we doubt that the students used as participants in study two, who were largely people who favor dogs over cats, would be aware that August 8 is World Cat Day and thus this contrast would likely have little, if any, affect on this particular sample of participants. A similar, but even more relevant, day in the United States, where the study was performed, is National Black Cat Day, which is celebrated on October 27. If the participants had been aware of World Cat Day, they would possible be aware of National Black Cat Day and this would likely have skewed the results in the opposite direction. We have added a brief paragraph in the future research section of the manuscript that addresses this idea.
From a practical standpoint, the participants in the second study were university students and the university was not in session on August 8. - I have no reservations about the statistical methods.
- I miss the reference in the conclusions to the application of the obtained results in everyday practice.
Response: Thank you for catching this oversight on our part. We have added a sentence that explains why being able to manipulation black cat bias might have the important practical consequence of reducing the length of stay of black cats in shelters.

Reviewer 3 Report
Comments and Suggestions for Authors
I think this is a relatively unique study, building on a similar piece of work conducted a number of years ago. I would say there needs to be some expansion on explaining why the study and the results are important, and how we can use them practically given the limitations, as well as ensuring that it is clear that these results are likely location specific and not widely generalizable. Some expansion of the methods is also required to fully understand how the data was analysed to ensure it is repeatable by others.
Generally:
Given the small sample size, I would frame the paper more clearly as being from a US perspective. I have made a few specific comments regarding this framing below.
40: The phrase ‘western societies’ is a little dated, I would be more specific and say the US, particularly as your examples of specific superstitions stem from there, or the US and Europe if you want to include a wider geographical range.
40: Would remove ‘There are many superstitions about black cats’ as reads as repetitive and just have the reference for the sentence before.
58-63: Would condense this and select just a few to highlight the point.
65-79: Highlight that this is country specific, e.g. data in the UK suggests black cats do not typically take significantly longer to adopt, and UK rescue centres do not typically euthanise healthy cats.
103: Include the reference after Jones and Hart again.
146-147: This makes an assumption that cat or dog people have cared for these animals as pets but might not be true, unless this was a criteria for participation?
176: Issue with bold text here
179: I’m not convinced this would reduce inter-cultural differences. If you have ethnicity information I would include it and discuss it, however if not I would suggest changing to ‘We recruited participants who reported that they were currently living in the United States of America and who were native speakers of English.’ only.
201: Why 52?
211: Did non-black cats have no black at all? Or did it include white and black cats, or tortoiseshells/calico etc.
240: I’m not sure what ‘an informal survey of students in another class’ means
245: Change ‘should’ to ‘may’, and explain why you think this briefly
258: Either omit or change to something along the lines of ‘The BPP scale as detail in study one were utilised’, as S1 also includes descriptions of the participants which were different here.
269-280, 304-306: The results should not contain any hypotheses/discussion, nor information on calculations which should be included in the methods; ensure this section simply reports the findings.
303: Include results of the BPP somewhere so readers can see generally how strongly participants scored, might be appropriate for the appendix.
312-319, 346-349: Ensure the methods section contains all of the information on the calculations carried out on the data, so that it is easily repeatable by other researchers.
359-362: Great detail to include, however would fit better in the discussion regarding study limitations.
364: Would provide a little more info after ‘black cat bias was found’ e.g. in cat/dog/dog and cat people, in which study etc.
367: What is your justification for avoiding them?
378-380: Think the mention of feline behaviour and interpretation of affective state is good, but why would this only confer bias in interpretation of photos of black cats, and not those of other colours?
382-395, 399-401: If is an assumption that cat-people actually had previously/did currently own a cat, make that clear here (same for dog and dog-and-cat people), if it was asked, include in the methods and results.
422: Recognise the welfare implications of collars if suggesting them, and recommend a quick-release.
418-429: Could it be that people simply find other colours of cats prettier? Rather than there being an inherent bias based on superstition or personality.
Discussion generally: Need some acknowledgement of limitations, e.g. very small sample sizes, limited age range, small geographical location, absence of data regarding religious background of participants.
Additionally, expansion of how the results can be used practically would be beneficial to improve the impact of the paper.
Author Response
I think this is a relatively unique study, building on a similar piece of work conducted a number of years ago. I would say there needs to be some expansion on explaining why the study and the results are important, and how we can use them practically given the limitations, as well as ensuring that it is clear that these results are likely location specific and not widely generalizable. Some expansion of the methods is also required to fully understand how the data was analysed to ensure it is repeatable by others.
Generally:
- Given the small sample size, I would frame the paper more clearly as being from a US perspective. I have made a few specific comments regarding this framing below.
Response: We agree that this is an important distinction and have made modifications as suggested. - 40: The phrase ‘western societies’ is a little dated, I would be more specific and say the US, particularly as your examples of specific superstitions stem from there, or the US and Europe if you want to include a wider geographical range.
Response: Changed - 40: Would remove ‘There are many superstitions about black cats’ as reads as repetitive and just have the reference for the sentence before.
Response: Thank you. We have modified the sentence to reflect that the superstitions about black cats can be good or bad across cultures. - 58-63: Would condense this and select just a few to highlight the point.
Response: Per your helpful comment, we have deleted four of the seven comments. - 65-79: Highlight that this is country specific, e.g. data in the UK suggests black cats do not typically take significantly longer to adopt, and UK rescue centres do not typically euthanise healthy cats.
Response: We have highlighted that most of the studies are from shelters in the United States and that the prevalence of “kill” shelters varies by country. - 103: Include the reference after Jones and Hart again.
Response: The manuscript has been updated to include the citation. - 146-147: This makes an assumption that cat or dog people have cared for these animals as pets but might not be true, unless this was a criteria for participation?
Response: Thank you for catching this. Because ownership was not a criteria for participation, we have modified the sentences to include those cat people who own cats and those dog people who do not own cats. - 176: Issue with bold text here
Response: This issue was not in the manuscript as submitted. It must have arisen during the editorial process and has been fixed. - 179: I’m not convinced this would reduce inter-cultural differences. If you have ethnicity information I would include it and discuss it, however if not I would suggest changing to ‘We recruited participants who reported that they were currently living in the United States of America and who were native speakers of English.’ only.
Response: Thank you. Because we did not collect information on ethnicity, we have deleted the statement “To reduce, but not eliminate, possible intercultural difference that might exist in black cat bias”
- 201: Why 52?
Response: We have clarified that 52 is approximately half of the 114 participants. The randomness in the random assignment of participants to the description vs no-description groups did not yield a perfect 57-57 split. - 211: Did non-black cats have no black at all? Or did it include white and black cats, or tortoiseshells/calico etc.
Response: Thank you. We have clarified that the pictures of black cats were of cats that were entirely black with no white and no orange and that the non-black cats had no large areas of black in their pictures. - 240: I’m not sure what ‘an informal survey of students in another class’ means
Response: We have clarified this section of the manuscript.
- 245: Change ‘should’ to ‘may’, and explain why you think this briefly
Response: We have made the suggested change. We have explained that, in the United States, Halloween is associated with witches and that black cats are sometimes presented as witches’ familiars in decorations. Being exposed to such decorations might reinforce any relation between black cat bias and witchcraft.
- 258: Either omit or change to something along the lines of ‘The BPP scale as detail in study one were utilised’, as S1 also includes descriptions of the participants which were different here.
Response: We have updated the materials section for study two even though the study one materials section does not include a description of the participants. - 269-280, 304-306: The results should not contain any hypotheses/discussion, nor information on calculations which should be included in the methods; ensure this section simply reports the findings.
Response: We respectfully disagree. Different disciplines seem to have different preferences for where the plan for statistical analysis is placed in a manuscript. Like the American Psychological Association (APA), we believe that including the hypothesis, the description of the statistical test used, and the results of the statistical test in the same place makes it easier for the reader to follow what is being done with the statistical analysis and why it is being done.
The instructions to authors for Animals states that the results section should “Provide a concise and precise description of the experimental results, their interpretation as well as the experimental conclusions that can be drawn.” Interpretation and conclusions are based on the hypotheses being tested. Thus, in our opinion, moving the hypotheses from the results would make the manuscript less clear as the reader would not know which hypothesis are being tested by which statistical analysis and how the conclusions relate to the hypotheses.
- 303: Include results of the BPP somewhere so readers can see generally how strongly participants scored, might be appropriate for the appendix.
Response: Tables containing descriptive statistics for the BPP have been included in the supplementary material and are mention in the new section 3.1.5.
- 312-319, 346-349: Ensure the methods section contains all of the information on the calculations carried out on the data, so that it is easily repeatable by other researchers.
Response: As in a previous comment, different disciplines seem to have different preferences for where this information belongs in a manuscript. Consistent with the APA, it is our opinion that separating the scoring of the data from the statistical analysis of the data requires the reader to remember scoring details when reading the analysis. This does not lead to clarity. The instruction to authors is ambiguous stating that the methods should be described with sufficient detail to allow for replication. It does not specify whether methods is the design of the study or the statistical analysis of the study or both. We believe that there is sufficient information to replicate the design of the study in the method section and sufficient information to replicate the analysis of the study in the results section. This way of separating design and analysis is the accepted way in some disciplines, such as psychology. - 359-362: Great detail to include, however would fit better in the discussion regarding study limitations.
Response: Thank you. We have moved the sentence to the discussion section and expanded it. - 364: Would provide a little more info after ‘black cat bias was found’ e.g. in cat/dog/dog and cat people, in which study etc.
Response: Thank you for this helpful suggestion. We have provided more details about when black cat bias was found for each study. - 367: What is your justification for avoiding them?
Response: The justification for avoiding Jones and Hart’s questions is given in the sentences that immediately follow the statement that the questions were avoided. - 378-380: Think the mention of feline behaviour and interpretation of affective state is good, but why would this only confer bias in interpretation of photos of black cats, and not those of other colours?
Response: We apologize that the logic was not clear. We have modified the sentence to more explicitly state that, in the given situation, cats of all coat colors should be treated the same and that black cat bias should not be expressed. - 382-395, 399-401: If is an assumption that cat-people actually had previously/did currently own a cat, make that clear here (same for dog and dog-and-cat people), if it was asked, include in the methods and results.
Response: The question is mentioned at line 188 in the original manuscript’s method section. The data are in the original manuscript at section 3.1.4. - 422: Recognise the welfare implications of collars if suggesting them, and recommend a quick-release.
Response: We have added “break-away” before “collar” to recognize the danger than other collars can pose to animals. - 418-429: Could it be that people simply find other colours of cats prettier? Rather than there being an inherent bias based on superstition or personality.
Response: Thank you for this idea. We have added it to the discussion. - Discussion generally: Need some acknowledgement of limitations, e.g. very small sample sizes, limited age range, small geographical location, absence of data regarding religious background of participants.
Response: In the discussion of the second study, we have added “The finding…needs to be interpreted with caution…due to the small sample of college students in an introductory psychology course, that was used.” Because we looked at religiosity as measured by the BPP and not religious background, we are not sure why the lack of information on religious background is a limitation. We have added that the results apply to people living in the United States to the first sentence of the discussion. - Additionally, expansion of how the results can be used practically would be beneficial to improve the impact of the paper.
Response: We have added a section to the discussion that discusses a possible practical application of the results. Given that the study did not directly address length of stay issues associated with black cats, it is questionable whether the results can be applied to that situation.

Round 2
Reviewer 2 Report
Comments and Suggestions for Authors
I thank the Authors for taking the trouble to respond to my concerns. I appreciate the diplomatic language in those cases where I probably did not understand the intention of the Authors precisely enough.
I am glad that some of my comments were well received and used in the text of the manuscript.
Indeed, the cultural difference between the countries of the old continent (Europe) and the United States may cause a different approach to issues related to human-animal interaction.
The corrections and additions made (not only related to my review) significantly improved the overall reception of the article.
Author Response
[Comment] The authors have done a good job of making the changes recommended by the reviewers, but they opted not to accept the recommendation of Reviewer 3 to add information about the analysis into the Methods section. I disagree with the authors' argument that reporting analyses only in the Results is common practice in psychology. It is not common practice in the field of psychology I research, even if it is recommended by APA - which, incidentally, is not the formatting system used in this journal. I do agree that having it in the Results section ALSO, as a reminder to the reader, is a good idea. However, I advise the authors to add an Analysis subsection to the end of the Methods with an overview of the hypotheses and how they were analysed, so that people who want to replicate the study in the future can do so solely by reading the Methods. Once done, I will be happy to accept this paper for publication.
[Response] Thank you. We have added the following sections to the method section:
2.1.4. Data Analysis
The BPP was scored following the method given in [26]. The implicit measure of black cat bias was created following the method given in [24]. The explicit measure of black cat bias was created by subtracting the mean responses to the questions “I would like to live with this cat” and “This is a good cat” for black cats from the corresponding mean responses for non-black cats.
For both explicit and implicit measures of black cat bias, one-tailed, one-sample t-tests were used to determine if black cat bias likely exists. To protect the family-wise α level, the Bonferroni correction was applied. To determine if superstitious behaviors, beliefs in witchcraft, and religiosity are predictors of implicit and explicit measures of black cat bias, Pearson’s rs were calculated. To determine if descriptions of cats reduced black cat bias, one-tailed, independent samples t-tests were used. A one-way, independent samples ANOVA with Tukey multiple comparisons was used to determine if the number of cats owned by cat-people, dog-people, and dog-and-cat-people differed.
2.2.4. Data Analysis
Implicit and explicit measures of black cat bias were scored as in study one. To determine if black cat bias existed far from and close to Halloween, one-tailed, t-tests were used for both implicit and explicit measures of black cat bias.